# SC2 Benchmark: Supervised Compression for Split Computing

**Yoshitomo Matsubara** *  
*Department of Computer Science*  
*University of California, Irvine*  
*yoshitom@uci.edu*

**Ruihan Yang**  
*Department of Computer Science*  
*University of California, Irvine*  
*ruihan.yang@uci.edu*

**Marco Levorato**  
*Department of Computer Science*  
*University of California, Irvine*  
*levorato@uci.edu*

**Stephan Mandt**  
*Departments of Computer Science and Statistics*  
*University of California, Irvine*  
*mandt@uci.edu*

**Reviewed on OpenReview:** *https://openreview.net/forum?id=p28wv4G65d*

## Abstract

With the increasing demand for deep learning models on mobile devices, splitting neural network computation between the device and a more powerful edge server has become an attractive solution. However, existing split computing approaches often underperform compared to a naive baseline of remote computation on compressed data. Recent studies propose learning compressed representations that contain more relevant information for supervised downstream tasks, showing improved tradeoffs between compressed data size and supervised performance. However, existing evaluation metrics only provide an incomplete picture of split computing. This study introduces *supervised compression* for split computing (SC2) and proposes new evaluation criteria: minimizing computation on the mobile device, minimizing transmitted data size, and maximizing model accuracy. We conduct a comprehensive benchmark study using 10 baseline methods, three computer vision tasks, and over 180 trained models, and discuss various aspects of SC2. We also release our code[1] and `sc2bench`,[2] a Python package for future research on SC2. Our proposed metrics and package will help researchers better understand the tradeoffs of supervised compression in split computing.

## 1 Introduction

Machine learning models are increasingly used in intelligent devices such as smart devices, wearable devices, autonomous drones, and surveillance cameras (Chen & Ran, 2019). However, these devices are often computationally weak, which makes it challenging to deploy complex deep learning models on them (Eshratifar et al., 2019a). To address this issue, researchers have proposed lightweight machine learning models that are optimized for low computational cost and high supervised performance (Sandler et al., 2018; Tan et al., 2019; Howard et al., 2019). An alternative approach is to offload heavy computing tasks to a more powerful cloud/edge server. In this scenario, weak local devices only send compressed data such as images to the cloud/edge server, which carries out heavy computing costs to run complex deep learning models. In the context of visual data, neural image compression models have been

---

*This work was done prior to joining Amazon.  
[1] https://github.com/yoshitomo-matsubara/sc2-benchmark  
[2] https://pypi.org/project/sc2bench/

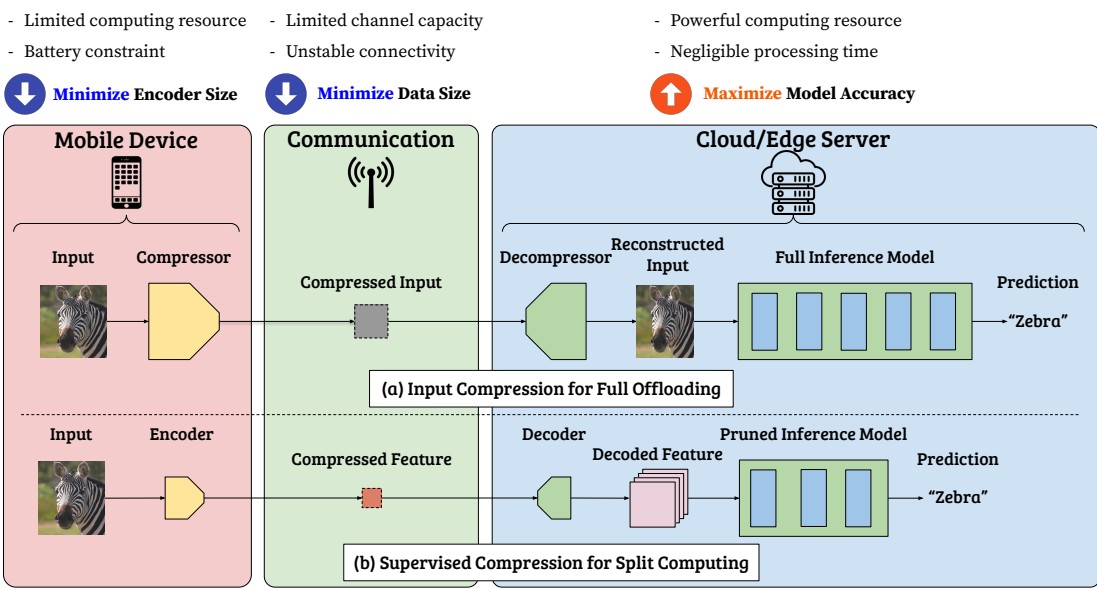

Figure 1: Input compression (**top**) vs. SC2: supervised compression for split computing (**bottom**) for image classification. The input compression approach reconstructs the input image on the could/edge server whereas the SC2 approach produces an compressible feature representation suitable for the supervised task. Note that the training process is done offline, and then the model will be split for deployment.

attracting much interest (Ballé et al., 2017; Minnen et al., 2018; Yang et al., 2023b). However, compressing the full image using traditional algorithms optimized for input reconstruction is inefficient, as they likely result in embedding information that is irrelevant toward the supervised task (Choi & Han, 2020).

A potentially better solution is to split the neural network model into two sequences (Kang et al., 2017). The first sequence is executed on the weak mobile (local) device and applies some elementary feature transformations to the input data. Then, the mobile device transmits intermediate, informative features through a constrained wireless communication channel to the edge server so that the second sequence of the model can processes the bulk part of the computation (Eshratifar et al., 2019b; Matsubara et al., 2019). This approach is called split computing, and it has been advancing with learnable data compression pipelines based on feature quantization (Matsubara et al., 2020; Shao & Zhang, 2020; Matsubara & Levorato, 2021; Banitalebi-Dehkordi et al., 2021; Assine et al., 2022), or entropy-coding (Singh et al., 2020; Matsubara et al., 2022c; Yuan et al., 2022).

In this paper, we formalize and study the problem of supervised compression for split computing (SC2). We benchmark several established methods, including state-of-the-art input and feature compression baselines, for various models in multiple supervised tasks from the aspects of the SC2 problem. Our experiments reveal that the choices of the splitting point, reference models, and the encoder-decoder architectures substantially affect the supervised rate-distortion performance. We argue that the study of these approaches necessitates a thorough characterization of the three-way tradeoff between encoder size, data size, and supervised learning performance.

This benchmark study involves 10 different state of the art methods and more than 180 trained models for three different computer vision tasks. It also contains a variety of architecture ablations, extended discussions on the limitations of the supervised rate-distortion tradeoff, and the first discussion of a three-way tradeoff in supervised compression. Our main contributions are proposing new metrics, an exhaustive benchmarking study, and providing a Python package and code repository[1] for future research in the research community.

The rest of this paper is structured as follows. In Section 2, we summarize related studies, supervised compression, and the motivations of this benchmark study. Section 3 provides an overview of SC2. In Section 4, we demonstrate the effectiveness of our approach, highlighting the importance of our proposed tradeoffs for SC2, and further discuss factors to improve SC2 Benchmark results. Finally, we conclude this work in Section 5.

## 2 Background

In this section, we briefly introduce related studies and highlight the main motivation behind this study.

### 2.1 Related Work

We summarize related ideas from the neural compression and split computing communities.

#### 2.1.1 Neural Image Compression

Neural image compression models are typically trained in an unsupervised manner to reconstruct input images while learning compressed representations of the data (Yang et al., 2023b). These models leverage neural networks for nonlinear dimensionality reduction and subsequent entropy coding. Early studies employed LSTM networks to capture spatial correlations among pixels within an image (Toderici et al., 2017; Johnston et al., 2018).

One of the pioneering works in image compression using autoencoders was proposed by Theis et al. (2017). The connection between image compression and probabilistic generative models was established by variational autoencoders (VAEs)(Kingma & Welling, 2014; Ballé et al., 2017). Building upon this, Ballé et al. (2018) proposed two-level VAE architectures with a scale hyper-prior for image encoding. These architectures have been further enhanced by incorporating autoregressive structures(Minnen et al., 2018; Minnen & Singh, 2020) and optimizing the encoding process (Yang et al., 2020a). This approach has been extended to numerous architectures for images (Cheng et al., 2020; Zhu et al., 2021; Wang et al., 2022; He et al., 2022; Liu et al., 2022; Yang & Mandt, 2022a) and video (Wu et al., 2018; Djelouah et al., 2019; Han et al., 2019; Chen et al., 2019; Lu et al., 2019; Habibian et al., 2019; Agustsson et al., 2020; Yang et al., 2023a). Active research topics include variable bitrates (Lu et al., 2021), compression without pre-defined quantization grids (Flamich et al., 2019; Yang et al., 2020b), and exploring compression limits (Alemi et al., 2017; Yang & Mandt, 2022b).

While self-supervised compression architectures for generic image classification have been proposed (Dubois et al., 2021), one particular approach using a Vision Transformer (ViT)-based encoder from the CLIP model (Radford et al., 2021) by Dubois et al. (2021) is characterized by an encoder with 87.8 million parameters. However, this high parameter count, which is 627 times larger than the encoder proposed by Matsubara et al. (2022c), mainly due to the ViT-based encoder, renders it unsuitable for deployment on resource-constrained edge computing systems.

#### 2.1.2 Split Computing

In many real-time application settings, local (mobile) devices capture sensor data (*e.g.*, images) and often have limited computing resources and battery constraints, thus fully offload computationally heavy tasks to more powerful cloud/edge servers. In such full offloading scenarios, the more resourceful edge server receives the sensor data from the mobile device via a wireless communication channel and then execute the entire model. To complete the inference in a timely manner, the latter strategy requires a high-capacity wireless communication channel between the mobile device and edge server. With low-capacity wireless networks, critical performance metrics such as end-to-end latency would degrade compared to local computing due to the large communication delay. As an intermediate option between local computing and full offloading (the full computation is on either local device or edge server), split computing (Kang et al., 2017) has been attracting considerable attention from the research community to minimize total delay in resource-limited networked systems (Eshratifar et al., 2019b; Matsubara et al., 2019). For instance, Long Range (LoRa) (Samie et al., 2016) is a widely used technology for resource-constrained Internet of Things devices and applications, which has a data rate of 37.5 Kbps due to duty cycle limitations (Adelantado et al., 2017).

In split computing, a deep learning model is split into two sequences. The first sequence of the model is executed on the mobile device. Having received the output of the first section via a wireless communication, the second sequence completes the inference on the edge server. A critical need is to reduce computational load on the mobile device while minimizing communication cost (data size) as processing time on the edge server is often smaller than local processing and communication delays (Matsubara & Levorato, 2021). In order to reduce communication cost, recent studies on split computing (Eshratifar et al., 2019b; Matsubara et al., 2019; Shao & Zhang, 2020; Matsubara & Levorato, 2021; Assine et al., 2022) introduce a *bottleneck*, whose data size is smaller than input sample to vision models. Those studies

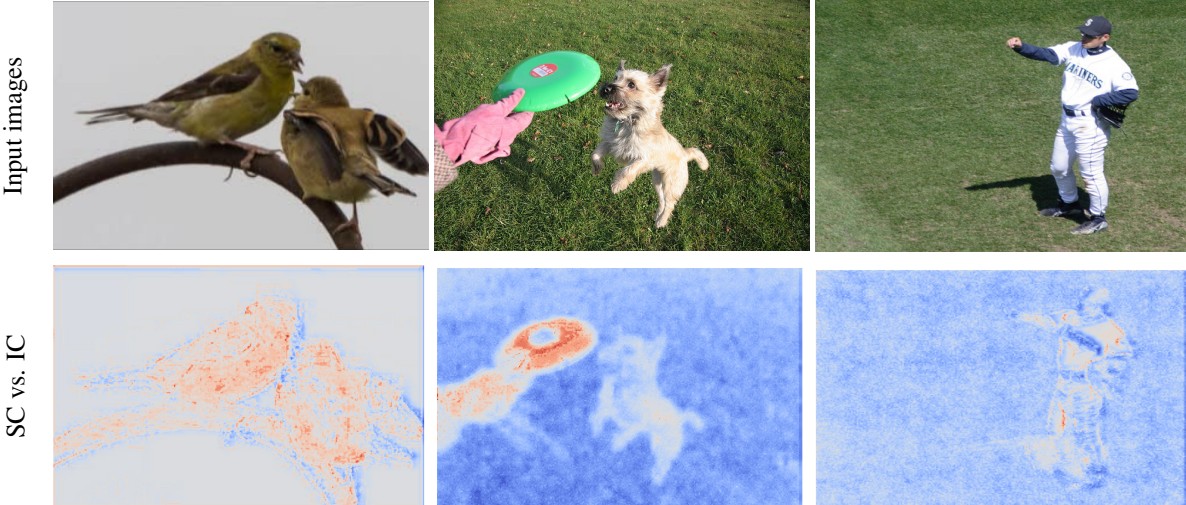

Figure 2: Bitrate comparison between a supervised compression (SC) model (Matsubara et al., 2022c) and an input compression (IC) model (Ballé et al., 2018). We plot the difference of the bits allocated for each pixel, exemplified on three images. Areas where the SC model allocates fewer and more bits for the given image are indicated in blue and red, respectively (best viewed in PDF). It is apparent how supervised compression allocates more bits to information relevant to the supervised object recognition goal.

show for image classification and object detection tasks that a combination of 1) channel reduction (dimensionality reduction) in convolution layers and 2) quantization at bottleneck layer is key to designing such bottlenecks.

## 2.2 Supervised Compression

Supervised compression refers to the process of learning compressed representations specifically tailored for supervised downstream tasks, including classification, detection, or segmentation. In this context, a supervised compression model is defined as a deterministic mapping $\mathbf{x} \mapsto \mathbf{z}^s \mapsto \mathbf{y}$, where $\mathbf{x}$, $\mathbf{y}$, and $\mathbf{z}^s$ represent the input data, targets, and compressed representations, respectively, for the given supervised downstream task(s).

This formulation bears resemblance to the concept of the deep variational information bottleneck (Alemi et al., 2017), although it should be noted that the latter was originally devised for enhancing adversarial robustness rather than compression. For comparison, we also consider a deterministic mapping $\mathbf{x} \mapsto \mathbf{z}^u \mapsto \hat{\mathbf{x}} \mapsto \mathbf{y}$ to denote the inference process of an input compression model followed by a supervised model. Here, $\mathbf{z}^u$ represents the compressed representations of the input data $\mathbf{x}$, while $\hat{\mathbf{x}}$ indicates the reconstructed input data.

It is important to highlight that in unsupervised input compression, the input data $\mathbf{x}$ can be accurately reconstructed from $\mathbf{z}^u$ ($\mathbf{x} \simeq \hat{\mathbf{x}}$). However, in the case of supervised compression, the compressed representations $\mathbf{z}^s$ lack the necessary information for a precise reconstruction of the original input data. This is because supervised compression aims to learn $\mathbf{z}^s$ in a way that retains relevant information for the specific downstream task(s), allowing for the compression of irrelevant information. For instance, in image classification tasks, not all pixels in input images are crucial, and supervised compression can effectively discard such irrelevant information in $\mathbf{z}^s$, while $\mathbf{z}^u$ requires all information to faithfully reconstruct the original input data $\mathbf{x}$.

Since input compression models are trained to reconstruct images, they tend to allocate bits more or less uniformly across the image. In contrast, supervised compression methods use the compressed features for prediction tasks; they should therefore be expected to allocate most of their bits to only *relevant* regions of the image, *i.e.*, regions that correlate with the prediction task. Figure 2 confirms this intuition, showing the difference in the bitrates between supervised compression and input (image) compression approaches. In more detail, we create spatial maps of the bits consumed for each pixel for an image compression model (Ballé et al., 2017) and a supervised compression model, Entropic Student (Matsubara et al., 2022c) and plot the differences (bottom row). Blue and red areas indicate

pixels for which the neural image compression model allocates more and fewer bits to each pixel than the supervised compression model does, respectively.

In this study, we put our focus on supervised compression for split computing (SC2) and present the details of our SC2 benchmark and experiments in Sections 3 and 4, respectively.

## 2.3 Motivations

In this section, we discuss the motivation behind the SC2 benchmark study.

### 2.3.1 Issues in Evaluation Metrics

As described in Section 2.1.2, split computing has been attracting a growing interest from the research community. Contributions from the machine learning community (Singh et al., 2020; Matsubara & Levorato, 2021; Matsubara et al., 2022c; Assine et al., 2022; Datta et al., 2022; Ahuja et al., 2023) specifically aim at improving the tradeoff between compressed data size and model accuracy. Many of such contributions leverage ideas and techniques widely used in neural image compression techniques such as reparameterization trick (Kingma & Welling, 2014), quantization with entropy coding Wintz (1972); Netravali & Limb (1980), and rate-distortion autoencoders (Ballé et al., 2017).

Such studies from the machine learning community still heavily rely on a rate-distortion evaluation metric that is popular in the neural image compression community (Ballé et al., 2017; 2018; Minnen et al., 2018; Minnen & Singh, 2020; Yang et al., 2020b;a; Yang & Mandt, 2023). However, the rate-distortion metric does not consider one of the essential aspects to determine the success of these techniques in real-world systems, that is, the cost of encoding given the heavily asymmetric computing power of mobile devices and edge servers. As emphasized in Section 2.1.2, it is important to reduce both the encoding cost (local computing cost on mobile devices) and size of data to be transferred from a weak mobile device to a powerful cloud/edge server.

For instance, Singh et al. (2020); Datta et al. (2022); Ahuja et al. (2023) discuss the supervised rate-distortion tradeoff for their approaches in the context of image classification, object detection, and/or semantic segmentation tasks. To achieve better supervised rate-distortion tradeoff, those studies introduce bottlenecks (splitting points) to existing convolution neural network models almost at the end of their layers – *e.g.*, the penultimate layers or last convolution layers/blocks in the original models, which results in approximately 60-170 times larger encoder size than encoder size of Entropic Student (Matsubara et al., 2022c) (see Section 4.3). Since those are discriminative models, it should be easier to compress data with respect to the input data (*e.g.*, images) when introducing bottlenecks at the later stage of the models. As a result, such approaches will put most of the model's inference cost on weak mobile devices to compress data, a strategy that increases total execution time and imposes high energy consumption.

The neural image compression community (Ballé et al., 2017; 2018; Minnen et al., 2018; Minnen & Singh, 2020; Yang et al., 2020b;a), Singh et al. (2020); Datta et al. (2022); Ahuja et al. (2023) refers to bits per pixel (BPP) as *rate* as part of the rate-distortion tradeoff evaluation. Conversely, split computing studies such as (Matsubara et al., 2019; Matsubara & Levorato, 2021; Assine et al., 2022; Haberer & Landsiedel, 2022), instead, focus on the actual data size of compressed representations. Reducing the data size directly decreases the data communication time between the mobile device and the edge server, while BPP does not so necessarily (*e.g.*, larger data size can result in small BPP if the image has more pixels). In order to improve the effectiveness of split computing at runtime, the performance evaluation should focus on the actual data size of compressed representations in SC2 problems.

To address those issues, in this study we define evaluation criteria for SC2 in Section 3.1 and propose how to incorporate them into existing rate-distortion tradeoffs in Section 3.2.

### 2.3.2 Lack of Comprehensive Benchmark

Besides the issues in evaluation metrics for SC2 (Section 2.3.1), we point out that a comprehensive discussion on many important aspects of SC2 is lacking in the machine learning community. We report some examples in the following:

1. ***Multiple target tasks***: Currently, supervised compression studies (Matsubara & Levorato, 2021; Singh et al., 2020; Matsubara et al., 2022c; Yuan et al., 2022) discuss the performance of their methods for different tasks, using different evaluation metrics. In order to highlight the differences between the proposed methods, it is

essential to discuss the performance on a shared set of tasks. *e.g.*, does one method consistently outperform other methods on all the tasks?

2. ***Bottleneck placement***: For efficient inference in split computing, it is critical to reduce both encoding cost and data size of compressed representations. Moreover, determining where to introduce bottlenecks in a model is a relevant question as it influences the allocation of computing load in the system. These aspects and tradeoffs should be discussed in state of the art supervised compression methods, also providing a comparison with image-codec-based feature compression baselines (Alvar & Bajić, 2021).

3. ***Variety of reference models***: Image classification models play an important role as they are used as backbones for complex computer vision tasks such as object detection and semantic segmentation (*e.g.*, ResNet (He et al., 2016) as a backbone for Faster R-CNN with FPN (Ren et al., 2015; Lin et al., 2017a) and DeepLabv3 (Chen et al., 2017b) respectively). There is a trend in both supervised compression and split computing studies (Eshratifar et al., 2019b; Matsubara et al., 2019; Matsubara & Levorato, 2021; Singh et al., 2020; Matsubara et al., 2022c; Yuan et al., 2022) introduce bottlenecks to some existing models (referred to as *reference models* in this study) rather than design new models with bottlenecks from scratch. Besides CNN models, there is an increasing interest in Vision Transformer (ViT) (Dosovitskiy et al., 2021) in the machine learning and computer vision communities. Thus, investigating the effect of reference model choice for SC2 should be an interest of the communities.

4. ***More sophisticated encoder-decoder***: Several studies from the neural image compression community (Ballé et al., 2018; Minnen et al., 2018; Minnen & Singh, 2020) show that sophisticated encoder-decoder architectures such as a hyperprior significantly outperform simpler encoder-decoder architectures like factorized prior (FP) (Ballé et al., 2018) in terms of rate-distortion tradeoff for image compression tasks (unsupervised compression). For supervised compression, Matsubara et al. (2022c) use a FP-based encoder-decoder architecture. Following the study, Yuan et al. (2022) propose a hyperprior-based architecture, but do not compare its performance to that of the FP-based encoder-decoder. In addition to the choice of reference models, it should be important to discuss the impact of encoder-decoder architectures in the context of the SC2 problem.

Addressing the issues in SC2 evaluation metrics (Section 2.3.1), we tackle each of them through experiments with strong input and feature compression baselines in Sections 4.1 - 4.5.

## 3 Evaluation and SC2 Benchmark

In this section, we describe our evaluation criteria in detail. This includes concise definitions of the supervised rate-distortion tradeoff, the proposed tradeoffs between rate, distortion, and computing load, as well as our selection of baselines. Experimental results will be presented in Section 4.

### 3.1 Evaluation Criteria

In split computing, the following three components are typically considered: a low-powered local (mobile) device, a capacity-constrained network, and an edge (cloud) server. The overall goal is to distribute a neural network $\mathcal{M}$ such that the first layers are deployed on the local device, and the remaining layers are deployed on the cloud/edge server. The portion of the model deployed on the mobile device is considered an "encoder" because it compresses the data into a representation suitable for transmission to the edge server. Given the asymmetric nature of the system, the complexity of the encoder should be minimized, while the remaining part of the model should comprehend most of the computing load, as the edge server is assumed to have a larger computing power compared to the mobile device. We remark that the training process is performed offline, and the split computing is executed at runtime only. Training split deep neural network (DNN) models across multiple devices (Gupta & Raskar, 2018) (or split learning (Vepakomma et al., 2018)) is a different problem and out of the scope of this study.

As described in Section 2.2, we define supervised compression as *learning compressed representations for supervised downstream tasks* such as classification, detection, or segmentation. The design should aim for three criteria: high supervised performance (low *distortion*), high compressibility (low *rate*), and minimal *encoder size* on the local device (low *computing load*). We discuss these three aspects in more detail.

### 3.1.1 Supervised Distortion

The target metric highly depends on the supervised task; in the simplest case, it could be a form of accuracy, *e.g.*, in classification. In this paper, we study three applications of supervised compression involving classification, object detection, and semantic segmentation. In these cases, we consider the supervised distortion to be accuracy, mean average precision (mAP), and mean intersection over union (mIoU), respectively. Note that compressing intermediate model representations typically requires a discretization step at the bottleneck layer. These supervised distortions are therefore computed after such intermediate discretizations.

### 3.1.2 Compressed Data Size (Rate)

Rate is defined as the average file size per datum after compression. In the neural compression literature (Ballé et al., 2017; 2018; Minnen et al., 2018; Yang et al., 2020b; Singh et al., 2020), most studies focus on bits per pixel (BPP), defined as the number of bits in the compressed representation divided by the input image size. In this paper, we report the rate on the basis of data points since this measures the actual amount of data sent to the edge server. Frameworks should penalize large amount of data transferred from the mobile device to the edge server. Notably, BPP does not penalize absolute amount of data, for instance when feeding higher resolution images to downstream models for achieving higher model accuracy (Touvron et al., 2019). The rate could either be directly related to the size of the raw feature representations rounded to a certain arithmetic precision or result from an additional entropy coding step.

### 3.1.3 Encoder Size

In addition to minimizing rate and distortion, it is also critical to minimize local processing cost as mobile devices usually have battery constraints and limited computing power (Matsubara et al., 2020; 2022c). To estimate the local processing cost, FLOPS (floating point operations per second) and MAC (multiply-accumulate) (Zhang et al., 2019) are often used. However, FLOPS is not a static value, and FLOP/MAC is not well-defined in practice.[3] As a simple proxy to the computing cost, we measure the number of model parameters, a static value that widely used to discuss model complexity (He et al., 2016; Huang et al., 2017; Devlin et al., 2019; Matsubara et al., 2022c). We define the encoder size $E_{size}$ as the total number of bits needed to represent the parameters of the encoder:

$$E_{size} = \sum_{i \in |\Theta|} \#\text{bits}(\Theta_i), \tag{1}$$

where $\Theta$ is a set of the encoder's parameters, and $\#\text{bits}(\cdot)$ indicates the number of bits for its input. For comparison, we also demonstrate the usage of encoder FLOPS in Appendix E.

## 3.2 Supervised R-D Tradeoff and Three-way Tradeoff

In lossy data compression (Yang et al., 2023b), one typically studies the tradeoff between the rate and distortion (R-D) of a compression scheme, *i.e.*, the quality degradation as a function of file size (Ballé et al., 2017; 2018; Minnen et al., 2018; Singh et al., 2020). In analogy to this, split computing approaches typically consider a *supervised* R-D tradeoffs (Matsubara et al., 2020; Matsubara & Levorato, 2021; Singh et al., 2020; Matsubara et al., 2022c) and replace the reconstruction distortion with the supervised distortion defined above.

The conventional supervised rate-distortion (R-D) tradeoff does not consider the encoder size. This approach is reasonable as long as models are compared with similar encoder sizes. However, without restrictions on the encoder size, the supervised compression problem becomes trivial. For example, in a $K$-class classification setup, a powerful encoder could simply carry out the classification task on the local device and only compress the label, leading to a cheap $-\log(K)$ bit representation of the transmitted features. If the bottleneck layer is deployed close to the output layer, similarly small compression costs can be achieved (Singh et al., 2020; Datta et al., 2022; Ahuja et al., 2023). However, Matsubara et al. (2022c) emphasize the importance of minimizing the encoder size for achieving efficient split computing.

To also take the encoder size into account, we propose to analyze the **three-way tradeoff** between encoder size, data size, and supervised distortion. Naturally, not all criteria can be simultaneously fulfilled: upon choosing a lighter

---

[3]*E.g.*, https://detectron2.readthedocs.io/en/latest/modules/fvcore.html#fvcore.nn.FlopCountAnalysis

Table 1: List of methods in this study. IC: Input compression, FC: Feature compression, SC: Supervised compression.

| Name (Acronym) | Type | Description |
|---|---|---|
| JPEG | IC | Standard lossy image codec |
| WebP | IC | Classical lossy image codec (Google) |
| BPG | IC | State-of-the-art classical lossy image codec (Bellard) |
| FP | IC | Factorized prior (Ballé et al., 2018) |
| SHP | IC | Scale hyperprior (Ballé et al., 2018) |
| MSHP | IC | Mean-scale hyperprior (Minnen et al., 2018) |
| JAHP | IC | Joint autoregressive hierarchical prior (Minnen et al., 2018) |
| JPEG | FC | JPEG-based intermediate feature compression (Alvar & Bajić, 2021) |
| WebP | FC | WebP-based intermediate feature compression (Alvar & Bajić, 2021) |
| CR + BQ | SC | Channel reduction & bottleneck quantization (Matsubara & Levorato, 2021) |
| Compressive Feature | SC | End-to-end learning of compresive feature (Singh et al., 2020; Yuan et al., 2022) |
| Entropic Student | SC | Multi-stage fine-tuning with neural compression and knowledge distillation (Matsubara et al., 2022c) |

weight encoder, we naturally have a less compressible bottleneck representation or a drop in accuracy. In Section 4, we visualize several such tradeoff curves. To simplify plotting and ease model selection, we also propose another tradeoff that takes encoder size and data size multiplicatively into account. We term this quantity **ExR-D tradeoff**, where we plot the distortion as a function of the product of encoder size and data size (see Section 3.1).

### 3.3 Baselines

In this section, we discuss relevant baselines and categorize them as either input compression, feature compression or supervised compression methods. All these baselines and the corresponding acronyms are summarized in Table 1.

#### 3.3.1 Input / Feature Compression Baselines

In conventional implementations of the edge computing paradigm for computer vision tasks, compressed images are transmitted to the edge server, where all the tasks are then executed. We consider seven baselines in this study referring to the "input compression" scenario that can be categorized into either codec-based or neural input compression.

**Classical Image Compression.** A first approach relies on using off-the-shelf classical image compressors. We evaluate each model's performance in terms of the rate-distortion curve by setting different quality values for three codec-based input compression methods: JPEG, WebP (Google), and BPG (Bellard). We use the implementations in Pillow[4] and investigate the rate-distortion (R-D) tradeoff for the combination of the codec and pretrained downstream models by tuning the quality parameter in the range of 10 to 100. Since BPG is not available in Pillow, our implementation follows (Bellard), and we use tune the quality parameter in the range of 0 to 50 to observe the R-D curve. We use the x265 encoder with 4:4:4 subsampling mode and 8-bit depth for YCbCr color space, following (Bégaint et al., 2020). We also introduce codec-based feature compression baselines (Alvar & Bajić, 2021) (see Section 4.3).

**Neural Image Compression.** As an alternative, we consider state of the art neural image compressors (Ballé et al., 2018; Minnen et al., 2018; Minnen & Singh, 2020) (see (Yang et al., 2023b) for a recent survey). We adopted the neural image compression models whose pretrained weights were available in CompressAI (Bégaint et al., 2020). These models mainly rely on variational autoencoder architectures and differ in terms of their entropy models (priors) that can have a large effect on the achievable code lengths. Without going into detail, these models are known under the names of "factorized prior" (Ballé et al., 2018), "scale hyperprior" (Ballé et al., 2018), "mean-scale hyperprior" (Minnen et al., 2018), and "joint autoregressive hierarchical prior" (Minnen et al., 2018).

#### 3.3.2 Supervised Compression Baselines

Another group of baseline models originates from frameworks prior to split computing. We broadly divide them into the following three categories.

[4]https://python-pillow.org/

**Channel Reduction + Bottleneck Quantization.** These split computing baselines (Matsubara et al., 2020; Shao & Zhang, 2020; Matsubara & Levorato, 2021; Dong et al., 2022) correspond to reducing the bottleneck data size with channel reduction and bottleneck quantization; we hence denote them as CR+BQ. These methods quantize 32-bit floating-point to 8-bit integers (Jacob et al., 2018). Matsubara et al. (2020); Matsubara & Levorato (2021) report that post-training bottleneck quantization did not lead to significant accuracy loss. Following (Matsubara & Levorato, 2021), we modify these pretrained models and introduce bottlenecks with a different number of output channels in a convolution layer to control the bottleneck data size. Using the original pretrained model as a teacher model, we train the bottleneck-injected model (student) by generalized head network distillation (GHND) and quantize the bottleneck after the training session.

**End-to-End Supervised Compression.** As an instantiation of information bottleneck framework (Alemi et al., 2017), Singh et al. (2020) first propose an end-to-end supervised compression method with an entropy bottleneck for image classification tasks, and Yuan et al. (2022) apply a similar idea to object detection tasks. Singh et al. (2020)'s approach focuses on a single task and introduces the compressible bottleneck to the penultimate layer. In the considered setting, such a design leads to an overwhelming workload allocated to mobile devices: for example, in terms of model parameters, about 92% of the ResNet-50 (He et al., 2016) parameters would be deployed on the, weaker, mobile device. To make this approach compatible with SC2 setting, we apply their approach to entropic student models without a teacher model. We find that compared to (Singh et al., 2020), having a stochastic bottleneck at an earlier layer (due to limited capacity of mobile devices) leads to a model that is harder to optimize (see Section 4.1).

**Entropic Student.** Our final baseline in this paper is Entropic Student (Matsubara et al., 2022c), a two-stage fine-tuning method that combines the concepts of neural image compression and knowledge distillation (Hinton et al., 2014). At the first stage of the fine-tuning, only the encoder-decoder in the student model is trained to mimic intermediate representations of its teacher model. The second stage of the method freezes the parameters of its encoder and fine-tune decoder and all the subsequent layers for the target tasks so that the encoder can be reused for other tasks.

For the end-to-end supervised compression and Entropic Student methods, we individually train the same model architectures (including its encoder-decoder) with each of the two training methods. Following (Matsubara et al., 2022c), we design the encoder with convolution and GDN (Ballé et al., 2016) layers followed by a quantizer described in Appendix B. Similarly, the corresponding decoder is designed with convolution and inversed GDN (IGDN) layers to have the output tensor shape match that of the first residual block in ResNet-50 (He et al., 2016). For image classification, the entire architecture of the model consists of the encoder and decoder followed by the last three residual blocks, average pooling, and fully-connected layers in ResNet-50. For object detection and semantic segmentation, we replace ResNet-50 in Faster R-CNN (Ren et al., 2015) and DeepLabv3 (Chen et al., 2017b) with the student model for image classification.

### 3.4 Choice of Datasets

We use image data with relatively high resolution, including ImageNet (ILSVRC 2012) (Russakovsky et al., 2015), COCO 2017 (Lin et al., 2014), and PASCAL VOC 2012 datasets (Everingham et al., 2012). As pointed out in (Matsubara et al., 2022b), split computing is mainly beneficial for supervised tasks involving high-resolution images *e.g.*, $224 \times 224$ pixels or larger. For small data samples, either local processing or full offloading often achieve better operating points in the three way tradeoff compared to split computing.[5]

### 3.5 Python Package - `sc2bench` -

To facilitate research on supervised compression for split computing (SC2), we publish an installable Python package named `sc2bench` (*i.e.*, `pip install sc2bench`) and scripts to reproduce the experimental results reported in this paper.[1] This Python package is built on PyTorch (Paszke et al., 2019) and torchdistill (Matsubara, 2021) for reproducible SC2 studies, using CompressAI (Bégaint et al., 2020) and PyTorch Image Models (Wightman, 2019) for neural compression modules/models and reference models, respectively. Our Python package offers various supervised compression models, modules and functions for further studies on SC2, and our repository provides the implementations of our baseline models and training methods, including weights of the models we trained in this study.

---

[5]*E.g.*, the average data sizes of $32 \times 32$ pixels images for MNIST (LeCun et al., 1998) (Gray scale) and CIFAR (Krizhevsky, 2009) (RGB) are only 0.966 KB and 1.79 KB, respectively.

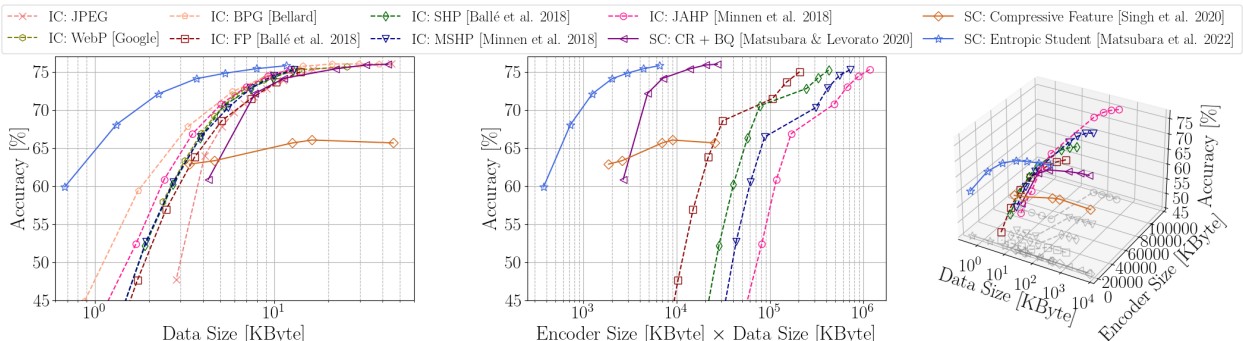

Figure 3: SC2 for image classification on ImageNet (ILSVRC 2012). We show the supervised R-D tradeoff (**left**), the ExR-D tradeoff (**middle**), and the full three-way tradeoff (**right**). In all cases, we used ResNet-50 as our reference model. Grey lines denote projections. Entropic Student performed best in R-D and ExR-D performance.

## 4 Experiments

We empirically assess state-of-the-art codec-based/neural input compression, feature compression, and supervised compression baselines (Table 1) according to the criteria introduced in Section 3. We thereby consider the three supervised tasks of image classification, object detection, and semantic segmentation.

### 4.1 Image Classification

We first discuss the rate-distortion performance of our baselines using a large-scale image classification dataset. Specifically, we use ImageNet (ILSVRC 2012) (Russakovsky et al., 2015), that consists of 1.28 million training and 50,000 validation samples. Using ResNet-50 (He et al., 2016) as a reference model, we train the models on the training split and report the top-1 accuracy on the validation split. Appendices C and D describe the details of the training configurations, including hyperparameters.

Figure 3 (left) presents supervised rate-distortion curves of ResNet-50 with various compression approaches, where the x- and y-axes show the expected compressed data size and the supervised performance (accuracy), respectively. For image compression baselines, we considered factorized prior (FP) (Ballé et al., 2018), scale hyperprior (SHP) (Ballé et al., 2018), mean-scale hyperprior (MSHP) (Minnen et al., 2018), and joint autoregressive hierarchical prior models (JAHP) (Minnen et al., 2018), as well as JPEG, WebP, and BPG codecs. The combination of channel reduction and bottleneck quantization (CR+BQ) (Matsubara & Levorato, 2021) – a popular approach used in split computing studies – is comparable to JPEG codec in terms of R-D tradeoff, but neural image compression models performed better.

Among all the baselines we considered, the Entropic Student trained by the two-stage method performed the best. To test the effect of knowledge distillation, we also trained the Entropic Student model without teacher model, which in essence corresponds to (Singh et al., 2020) with an adjusted bottleneck placement. The resulting R-D curve is significantly worse, which we attribute to two possible effects: first, it is widely acknowledged that knowledge distillation generally finds solutions that generalize better. Second, having a bottleneck at an earlier layer may make it difficult for the end-to-end training approach without a (pretrained) teacher model to optimize as empirically shown in (Matsubara et al., 2020; 2022a). We explore the effects of the bottleneck placement in Section 4.3.

Besides the tradeoff between data size and accuracy, we also investigated tradeoffs incorporating the encoder's model size. Figure 3 (middle) shows the proposed ExR-D tradeoff, and the full three-dimensional tradeoff is shown in Fig. 3 (right). Both the figures reveal that when considering encoder size, Entropic Student outperformed the other baselines even more significantly since the encoders of the split DNN models are significantly smaller than those in the neural input compression models. Entropic Student's encoder is approximately 40 times smaller than the encoder of the mean-scale hyperprior and can therefore be deployed efficiently on mobile devices. As shown in (Matsubara et al., 2022c), the model also can achieve a much shorter latency to complete the input-to-prediction pipeline (Fig. 1) than the input compression baselines we considered for resource-constrained edge computing systems.

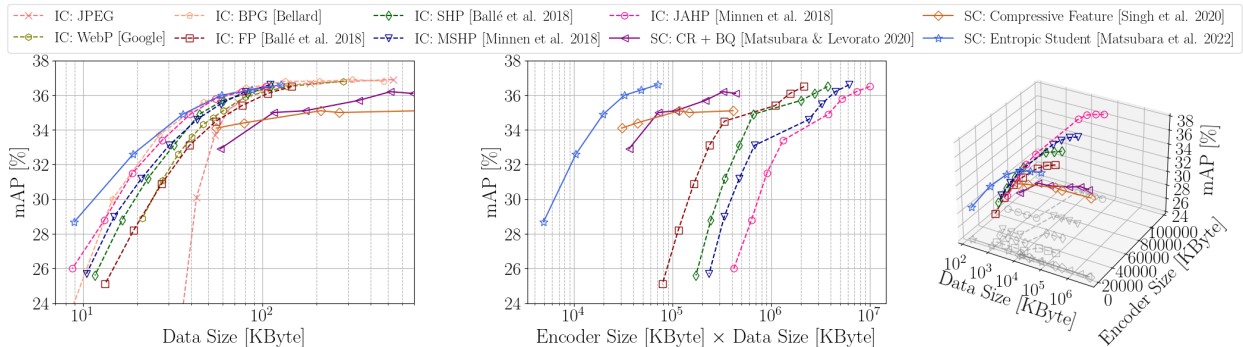

Figure 4: SC2 for object detection on COCO 2017. We show the supervised R-D tradeoff (**left**), the ExR-D tradeoff (**middle**), and the full three-way tradeoff (**right**). In all cases, we used Faster R-CNN with ResNet-50 and FPN as our reference model. Grey lines denote projections. Entropic Student performed best in R-D and ExR-D performance.

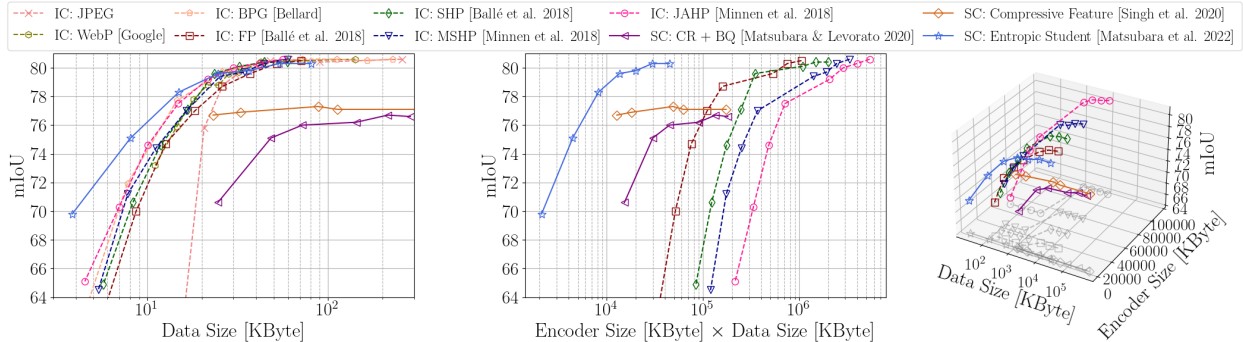

Figure 5: SC2 for semantic segmentation on PASCAL VOC 2012. We show the supervised R-D tradeoff (**left**), the ExR-D tradeoff (**middle**), and the full three-way tradeoff (**right**). In all cases, we used DeepLabv3 with ResNet-50 as our reference model. Grey lines denote projections. Entropic Student performed best in R-D and ExR-D performance.

## 4.2  Object Detection and Semantic Segmentation

We further study the rate-distortion performance on two downstream tasks: object detection and semantic segmentation, reusing the models pretrained on the ImageNet dataset. As suggested by He et al. (2019), such pre-training speeds up the convergence for other tasks (Russakovsky et al., 2015). Specifically, we train Faster R-CNN with FPN (Ren et al., 2015; Lin et al., 2017a) and DeepLabv3 (Chen et al., 2017b) for object detection and semantic segmentation, respectively, using the models pre-trained on ImageNet in Section 4.1 for the supervised compression baselines. Faster R-CNN is a two-stage object detection model; it generates region proposals and classifies objects in the proposed regions. DeepLabv3 is a popular semantic segmentation model (Chen et al., 2017a).

For object detection, we use the COCO 2017 dataset (Lin et al., 2014) to fine-tune the models. The training and validation splits in the COCO 2017 dataset have 118,287 and 5,000 annotated images, respectively. For detection performance, we refer to mean average precision (mAP) for bounding box (BBox) outputs with different Intersection-over-Unions (IoU) thresholds from 0.5 and 0.95 on the validation split. For semantic segmentation, we use the PASCAL VOC 2012 dataset (Everingham et al., 2012) with 1,464 and 1,449 samples for training and validation splits, respectively. We measure the performance by pixel IoU averaged over its 21 classes. It is worth noting that following the PyTorch (Paszke et al., 2019) implementations, the input image scales for Faster R-CNN (Ren et al., 2015) are defined by the shorter image side and set to 800 in this study which is much larger than the input image in the previous image classification task. For DeepLabv3 (Chen et al., 2017b), we use the resized input images such that their shorter size is 513. The training setup and hyperparameters used to fine-tune the models are described in Appendices C and D.

Figures 4 and 5 show the results for object detection and semantic segmentation, where the left figure shows the supervised R-D tradeoff. Compared to various split computing and input compression approaches, the Entropic Student approach demonstrates better supervised R-D curves in both the tasks. In object detection, Entropic Student's im-

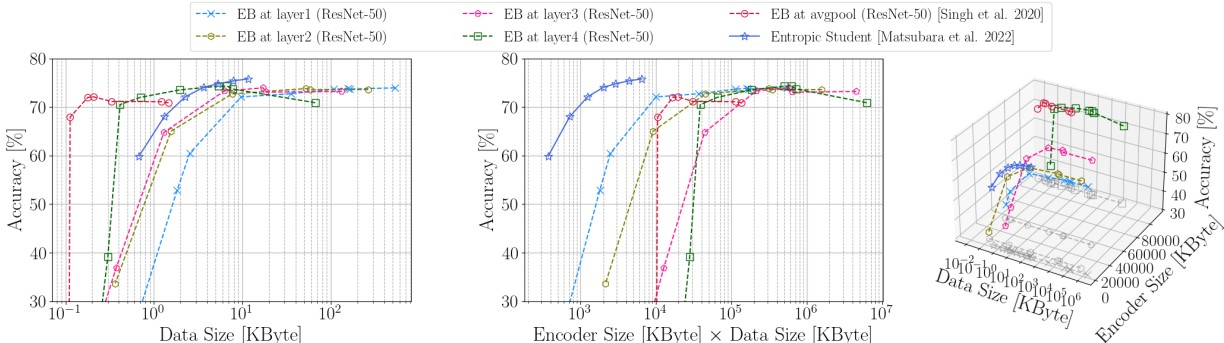

Figure 6: Entropic Student vs. ResNet-50 with entropy bottleneck (EB) introduced to different layers. Simply introducing EBs to its late layers *e.g.*, `layer4` and `avgpool` improved the conventional R-D tradeoff (**left**), which results in most of the layers in the model to be deployed on weak local devices. Our proposed ExR-D and three-way tradeoffs penalize such configurations (**middle** and **right**).

provements over BPG and JAHP are smaller than those in the image classification and semantic segmentation tasks. However, as shown in Figs. 4 and 5 (middle and right), the proposed ExR-D and three-way tradeoffs show that the combined improvements in encoder size and data size reductions are even more significant.

## 4.3 Bottleneck Placement

An important design decision in split computing is the choice of the layer where the DNN is split. If the DNN is split at an early layer, the computation on the weak local device will be lightweight, but the learned representation at the splitting point is not easily compressible. When splitting the DNN close to the output, the latent representation contains only relevant information for the supervised task and is therefore compressible, but most of the computation is carried out on the weak local device.

The proposed ExR-D tradeoff considers both the encoder size and the data size of the latent representation. It can guide selecting a bottleneck layer that simultaneously leads to a compressible representation and a lightweight encoder model. Figure 6 shows different R-D tradeoff curves, where we consider five ResNet-50 models that have entropy bottlenecks (EBs) placed at five different layers. Specifically, we introduce the EBs after the 1st, 2nd, 3rd, 4th residual blocks as well as at the penultimate layer; we refer to these models as `layer1`, `layer2`, `layer3`, `layer4`, and `avgpool` respectively. Note that Singh et al. (2020) introduce an EB at the penultimate layer of ResNet, and Datta et al. (2022); Ahuja et al. (2023) introduce an EB at `layer3` and `layer4` of ResNet-50, which results in deploying approximately 60-170 larger encoder than that of Entropic Student (Matsubara et al., 2022c) on weak mobile devices. We introduce the EB to a pretrained ResNet-50 and then fine-tune the model for better performance. For reference, we also add the entropic student model to the plot as it performed the best in Section 4.1.

Figure 6 (left) shows that the supervised R-D tradeoff can be misleading for choosing the bottleneck layer, simply favoring models that place their bottlenecks at late layers. With such configurations, we would deploy at least 92% of the entire model on the weak local device. Clearly, in split computing scenarios, we would rather prefer doing most of the computation on the edge server. A complete picture is seen on the ExR-D tradeoff (middle) and the three-way tradeoff (right), revealing that `layer1` would be the best placement for introducing bottlenecks (as done in the entropic student). These findings are not specific to learned entropy models but apply to conventional feature compression schemes as well. To demonstrate this, we adopt the method of (Alvar & Bajić, 2021) that used conventional image codecs to compress feature maps of a *pretrained* neural network. (Overall, this baseline leads to much worse compression rates, which is why we excluded it in our main experiments.) Thus, we implement a similar baseline by replacing the EB in a *pretrained* ResNet-50 with JPEG and WebP codecs. To be specific, we treat the network feature maps as a concatenation of 3-channel "sub-images", which can be compressed separately. In analogy to Fig. 6, we study the same bottleneck placements. Figure 7 shows the R-D, ExR-D, and three-way tradeoffs of the corresponding method, mirroring the findings of the previous discussion.

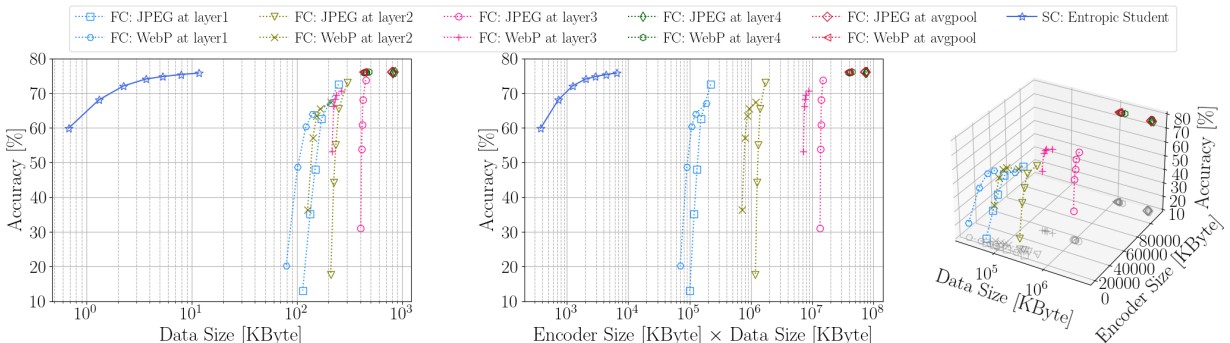

Figure 7: Entropic Student vs. ResNet-50 with codec-based feature compression approaches (instead of EB in Fig. 6) introduced to different layers. Overall, their data sizes are orders of magnitude larger than those of the supervised compression in Fig. 6.

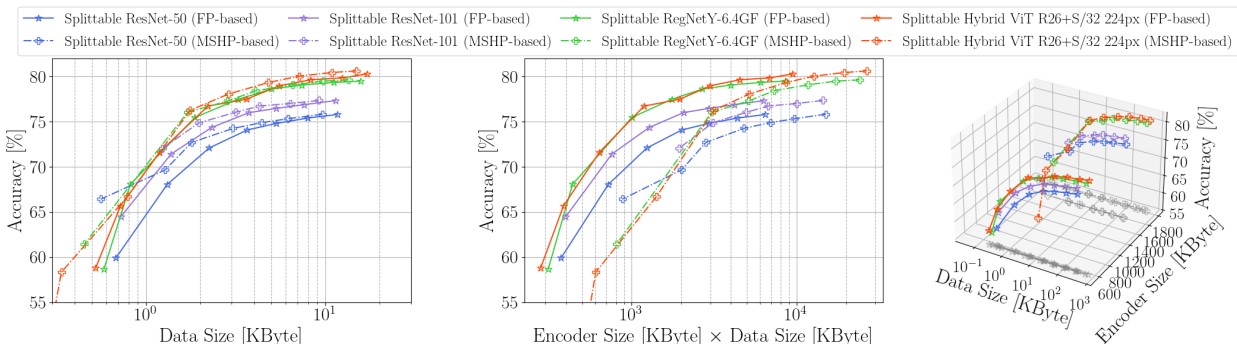

Figure 8: Conventional R-D (**left**), our proposed ExR-D (**middle**), and three-way tradeoffs (**right**) for Entropic Student with various reference models and encoder-decoder. Different colors indicate different reference models. Stronger reference models outperform Splittable ResNet-50, the best Entropic Student model in Fig. 3 in terms of R-D, ExR-D, and three-way tradeoffs. The encoder-decoder design well generalizes so that can work with various reference models including Hybrid ViT (Steiner et al., 2022), a transformer-based model. The more sophisticated (MSHP-based) encoder-decoder design further improved R-D tradeoff at cost of significantly increased encoder size.

## 4.4   Network Architecture Ablations

To further improve the performance of models for split computing without increasing the encoder size or data size, we put our focus on Entropic Student and investigate the effect of reference models. This section investigates alternatives to ResNet-50 as our default reference model and in particular considers ResNet-101 (a larger model in the family) (He et al., 2016), RegNetY-6.4GF (Radosavovic et al., 2020), and Hybrid Vision Transformer (ViT) R26+S/32 (224px) (Steiner et al., 2022) as reference models. All these models use the same input patch size of $224 \times 224$ and the same encoder-decoder architecture.

As shown in Fig. 8 (left), we significantly improved the R-D tradeoff by replacing ResNet-50 with the more accurate reference models. Since we reused the encoder-decoder design in Section 3.3.2 for the configurations *i.e.*, comparable encoder size and data size, we successfully improved the ExR-D and three-way tradeoffs as shown in Fig. 8 (middle and right). As most of the layers are deployed on the edge server (which we assume is not resource-constrained), using such more accurate reference models does not significantly affect the architecture's latency. Therefore, we can improve the ExR-D and three-way tradeoffs as long as the encoder-decoder can still learn suitable feature representations.

## 4.5   Supervised Compression with a Hyperprior

In this section, we discuss whether we can improve the supervised R-D tradeoff by adopting a more powerful entropy model for compressing the latent variables **z** losslessly. To this end, we draw on the neural compression literature,

where we adopt a hierarchical design in which the latent representations $\mathbf{z}$ are compressed using an entropy model that relies on additional *hyperlatents* $\mathbf{z}_h$. While the supervised encoder-decoder design in Entropic Student (see Section 3.3.2) is based on the factorized prior (FP) model of image compression (Ballé et al., 2018), the hyperprior approach draws on the mean-scale hyperprior (MSHP) model from (Minnen et al., 2018). The hyperprior can be expected to reduce the data size since the distribution of $\mathbf{z}$ can be better approximated.

In Fig. 8 (left), we confirm that the MSHP-based encoder-decoder consistently improved the R-D tradeoff compared to those with FP-based encoder-decoder. These gains are more significant for simpler reference models *i.e.*, ResNet-50 and -101. We stress that this extra performance comes at the cost of additional encoder complexity, which is reflected in the ExR-D curve that considers data size and encoder size jointly.

## 5 Conclusion

In this paper, we investigated supervised compression for split computing (SC2), where a machine learning model is split between a low-powered device and a much more powerful edge server. We explored optimal ways of splitting the network while aiming for high supervised performance, high compression rates, and low computational costs on the edge server. We introduced SC2 Benchmark, a new benchmarking framework of supervised compression for split computing to more rigorously analyze this setting and the various tradeoffs involved. Specifically, we investigated a variety of input/feature compression models, supervised compression models, supervised tasks (such as image classification, object detection, and semantic segmentation), training schemes (such as knowledge distillation), metrics (such as ExR-D and three-way tradeoffs), and architectures (convolutional or Vision Transformers). Altogether, this study involved more than 180 trained models. We showed that Entropic Student (Matsubara et al., 2022c), a supervised compression model inspired by neural image compression models (with or without hyperpriors), trained in a multi-stage knowledge distillation approach, performed best in terms of the supervised R-D, ExR-D, and three-way tradeoffs for the three supervised tasks considered in this study. Hoping that our benchmark will set the stage for a more rigorous evaluation of supervised compression methods and split computing models, we publish `sc2bench`, a pip-installable Python package to lower the barrier to SC2 studies. We also release our code repository[1] based on `sc2bench` to offer reproducibility of the experimental results in this study.

## Acknowledgments

We acknowledge the support by the National Science Foundation under the NSF CAREER award 2047418 and Grants 1928718, 2003237, 2007719, IIS-1724331 and MLWiNS-2003237, the Department of Energy, Office of Science under grant DESC0022331, the IARPA WRIVA program, as well as Disney, Intel, and Qualcomm.

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

# A   Image Compression Codecs

As image compression baselines, we use JPEG, WebP (Google), and BPG (Bellard). For JPEG and WebP, we follow the implementations in Pillow[6] and investigate the rate-distortion (RD) tradeoff for the combination of the codec and pretrained downstream models by tuning the quality parameter in range of 10 to 100. Since BPG is not available in Pillow, our implementation follows (Bellard) and we tune the quality parameter in range of 0 to 50 to observe the RD curve. We use the x265 encoder with 4:4:4 subsampling mode and 8-bit depth for YCbCr color space, following (Bégaint et al., 2020).

# B   Quantization

This section briefly introduces the quantization technique used in both proposed methods and neural baselines with entropy coding.

## B.1   Encoder and Decoder Optimization

As entropy coding requires discrete symbols, we leverage the method that is firstly proposed in (Ballé et al., 2017) to learn a discrete latent variable. During the training stage, the quantization is simulated with a uniform noise to enable gradient-based optimization:

$$\mathbf{z} = f_\theta(\mathbf{x}) + \mathcal{U}(-\frac{1}{2}, \frac{1}{2}). \tag{2}$$

At runtime, we round the encoder output to the nearest integer for entropy coding and the input of the decoder:

$$\mathbf{z} = \lfloor f_\theta(\mathbf{x}) \rceil. \tag{3}$$

## B.2   Prior Optimization

For entropy coding, a prior that can precisely fit the distribution of the latent variable reduces the bitrate. However, the prior distributions such as Gaussian and Logistic distributions are continuous, which is not directly compatible with discrete latent variables. Instead, we use the cumulative of a continuous distribution to approximate the probability mass of a discrete distribution (Ballé et al., 2017):

$$P(\mathbf{z}) = \int_{\mathbf{z}-\frac{1}{2}}^{\mathbf{z}+\frac{1}{2}} p(t)\mathrm{d}t, \tag{4}$$

where $p$ is the prior distribution we choose, and $P(\mathbf{z})$ is the corresponding probability mass under the discrete distribution $P$. The integral can easily be computed with the cumulative distribution function of the continuous distribution.

# C   Channel Reduction and Bottleneck Quantization

A combination of channel reduction and bottleneck quantization (CR + BQ) is a popular approach in studies on split computing (Eshratifar et al., 2019b; Matsubara et al., 2020; Shao & Zhang, 2020; Matsubara & Levorato, 2021; Choi et al., 2020; Dong et al., 2022), and we refer to the approach as a baseline.

## C.1   Network Architecture

### C.1.1   Image classification

We reuse the architectures of encoder and decoder from Matsubara and Levorato (Matsubara & Levorato, 2021) introduced in ResNet (He et al., 2016) and validated on the ImageNet (ILSVRC 2012) dataset (Russakovsky et al., 2015). Following the study, we explore the rate-distortion (RD) tradeoff by varying the number of channels in a convolution layer (2, 3, 6, 9, and 12 channels) placed at the end of the encoder and apply a quantization technique (32-bit floating point to 8-bit integer) (Jacob et al., 2018) to the bottleneck after the training session.

---

[6]https://python-pillow.org/

### C.1.2   Object detection and semantic segmentation

Similarly, we reuse the encoder-decoder architecture used as ResNet-based backbone in Faster R-CNN (Ren et al., 2015) and Mask R-CNN (He et al., 2017) for split computing (Matsubara & Levorato, 2021). The same ResNet-based backbone is used for Faster R-CNN (Ren et al., 2015) and DeepLabv3 (Chen et al., 2017b). Again, we examine the RD tradeoff by controlling the number of channels in a bottleneck layer (1, 2, 3, 6, and 9 channels) and apply the same post-training quantization technique (Jacob et al., 2018) to the bottleneck.

## C.2   Training

Using ResNet-50 (He et al., 2016) pretrained on the ImageNet dataset as a teacher model, we train the encoder-decoder introduced to a copy of the teacher model, that is treated as a student model for image classification. We apply the generalized head network distillation (GHND) (Matsubara & Levorato, 2021) to the introduced encoder-decoder in the student model. The model is trained on the ImageNet dataset to mimic the intermediate features from the last three residual blocks in the teacher (ResNet-50) by minimizing the sum of squared error losses. Using the Adam optimizer (Kingma & Ba, 2015), we train the student model on the ImageNet dataset for 20 epochs with the batch size of 32. The initial learning rate is set to $10^{-3}$ and reduced by a factor of 10 at the end of the 5th, 10th, and 15th epochs.

Similarly, we use ResNet-50 models in Faster R-CNN with FPN pretrained on COCO 2017 dataset (Lin et al., 2014) and DeepLabv3 pretrained on PASCAL VOC 2012 dataset (Everingham et al., 2012) as teachers, and apply the GHND to the students for the same dataset. The training objective, initial learning rate, and number of training epochs are the same as those for the classification task. We set the training batch size to 2 and 8 for object detection and semantic segmentation tasks, respectively. The learning rate is reduced by a factor of 10 at the end of the 5th and 15th epochs.

## D   Entropic Student

This section presents the details of end-to-end and multi-stage fine-tuning supervised compression baselines for Entropic Student models. We refer readers to (Matsubara et al., 2022c) for the architectures of Entropic Student models.

## D.1   Two-stage Training

Here, we describe the two-stage method proposed to train the Entropic Student models in (Matsubara et al., 2022c).

### D.1.1   Image classification

Using the ImageNet dataset, we put our focus on the introduced encoder and decoder at the first stage of training and then freeze the encoder to fine-tune all the subsequent layers at the second stage for the target task. At the 1st stage, we train the student model for 10 epochs to mimic the behavior of the first residual block in the teacher model (pretrained ResNet-50) in a similar way to (Matsubara & Levorato, 2021) but with the rate term to learn a prior for entropy coding. We use Adam optimizer with batch size of 64 and an initial learning rate of $10^{-3}$. The learning rate is decreased by a factor of 10 after the end of the 5th and 8th epochs.

Once we finish the 1st stage, we fix the parameters of the encoder that has learnt compressed features at the 1st stage and fine-tune all the other modules, including the decoder for the target task. By freezing the encoder's parameters, we can reuse the encoder for different tasks. The rest of the layers can be optimized to adopt the compressible features for the target task. Note that once the encoder is frozen, we also no longer optimize both the prior and encoder, which means we can directly use *rounding* to quantize the latent variable. With the encoder frozen, we apply a standard knowledge distillation technique (Hinton et al., 2014) to achieve better model accuracy, and the concrete training objective is formulated as follows:

$$\mathcal{L} = \alpha \cdot \mathcal{L}_{\mathrm{cls}}(\hat{\mathbf{y}}, \mathbf{y}) + (1 - \alpha) \cdot \tau^2 \cdot \mathcal{L}_{\mathrm{KL}}\left(\mathbf{o}^{\mathrm{S}}, \mathbf{o}^{\mathrm{T}}\right), \tag{5}$$

where $\mathcal{L}_{\mathrm{cls}}$ is a standard cross entropy. $\hat{\mathbf{y}}$ indicates the model's estimated class probabilities, and $\mathbf{y}$ is the annotated object category. $\alpha$ and $\tau$ are both hyperparameters, and $\mathcal{L}_{\mathrm{KL}}$ is the Kullback-Leibler divergence. $\mathbf{o}^{\mathrm{S}}$ and $\mathbf{o}^{\mathrm{T}}$ represent the *softened* output distributions from student and teacher models, respectively. Specifically, $\mathbf{o}^{\mathrm{S}} = [o_1^{\mathrm{S}}, o_2^{\mathrm{S}}, \ldots, o_{|\mathcal{C}|}^{\mathrm{S}}]$

where $\mathcal{C}$ is a set of object categories considered in target task. $o_i^{\text{S}}$ indicates the student model's softened output value (scalar) for the $i$-th object category:

$$o_i^{\text{S}} = \frac{\exp\left(\frac{v_i}{\tau}\right)}{\sum_{k \in \mathcal{C}} \exp\left(\frac{v_k}{\tau}\right)}, \tag{6}$$

where $\tau$ is a hyperparameter defined in Eq. 5 and called *temperature*. $v_i$ denotes a logit value for the $i$-th object category. The same rules are applied to $\mathbf{o}^{\text{T}}$ for teacher model, which is a target distribution.

For the 2nd stage, we use the stochastic gradient descent (SGD) optimizer with an initial learning rate of $10^{-3}$, momentum of 0.9, and weight decay of $5 \times 10^{-4}$. We reduce the learning rate by a factor of 10 after the end of the 5th epoch, and the training batch size is set to 128. The balancing weight $\alpha$ and temperature $\tau$ for knowledge distillation are set to 0.5 and 1, respectively.

### D.1.2 Object detection

We reuse the entropic student model trained on the ImageNet dataset in place of ResNet-50 in Faster R-CNN (Ren et al., 2015) and DeepLabv3 (Chen et al., 2017b) (teacher models). Note that we freeze the parameters of the encoder trained on the ImageNet dataset, following (Matsubara et al., 2022c). Reusing the encoder trained on the ImageNet dataset is a reasonable approach as 1) the ImageNet dataset contains a larger number of training samples (approximately 10 times more) than those in the COCO 2017 dataset (Lin et al., 2014); 2) models using an image classifier as their backbone frequently reuse model weights trained on the ImageNet dataset (Ren et al., 2015; Lin et al., 2017b).

To adapt the encoder for object detection, we train the decoder for 3 epochs at the 1st stage in the same way we train those for image classification (but with the encoder frozen). The optimizer is Adam (Kingma & Ba, 2015), and the training batch size is 6. The initial learning rate is set to $10^{-3}$ and reduced to $10^{-4}$ after the first 2 epochs. At the 2nd stage, we fine-tune the whole model except its encoder for 2 epochs by the SGD optimizer with learning rates of $10^{-3}$ and $10^{-4}$ for the 1st and 2nd epochs, respectively. We set the training batch size to 6 and follow the training objective in (Ren et al., 2015), which is a combination of bounding box regression, objectness, and object classification losses.

### D.1.3 Semantic segmentation

For semantic segmentation, we train DeepLabv3 in a similar way. At the 1st stage, we freeze the encoder and train the decoder for 40 epochs, using Adam optimizer with batch size of 16. The initial learning rate is $10^{-3}$ and decreased to $10^{-4}$ and $10^{-5}$ after the first 30 and 35 epochs, respectively. At the 2nd stage, we train the entire model except for its encoder for 5 epochs. We minimize a standard cross entropy loss, using the SGD optimizer. The initial learning rates for the body and the sub-branch (auxiliary module)[7] are $2.5 \times 10^{-3}$ and $2.5 \times 10^{-2}$, respectively. Following (Chen et al., 2017b), we reduce the learning rate after each iteration as follows:

$$lr = lr_0 \times \left(1 - \frac{N_{\text{iter}}}{N_{\text{max\_iter}}}\right)^{0.9}, \tag{7}$$

where $lr_0$ is the initial learning rate. $N_{\text{iter}}$ and $N_{\text{max\_iter}}$ indicate the accumulated number of iterations and the total number of iterations, respectively.

## D.2 End-to-end Training

In this work, the end-to-end training approach for learning compressive feature (Singh et al., 2020; Yuan et al., 2022)[8] is treated as a baseline and applied to the entropic student models without teacher models.

### D.2.1 Image classification

Following the end-to-end training approach (Singh et al., 2020), we train the entropic student model from scratch. Specifically, we use Adam (Kingma & Ba, 2015) optimizer and cosine decay learning rate schedule (Loshchilov & Hutter, 2017) with an initial learning rate of $10^{-3}$ and weight decay of $10^{-4}$. Based on their training objectives (Eq. 8),

---

[7] https://github.com/pytorch/vision/tree/main/references/segmentation
[8] Singh et al. (2020) and Yuan et al. (2022) assess their methods only for image classification and object detection tasks, respectively.

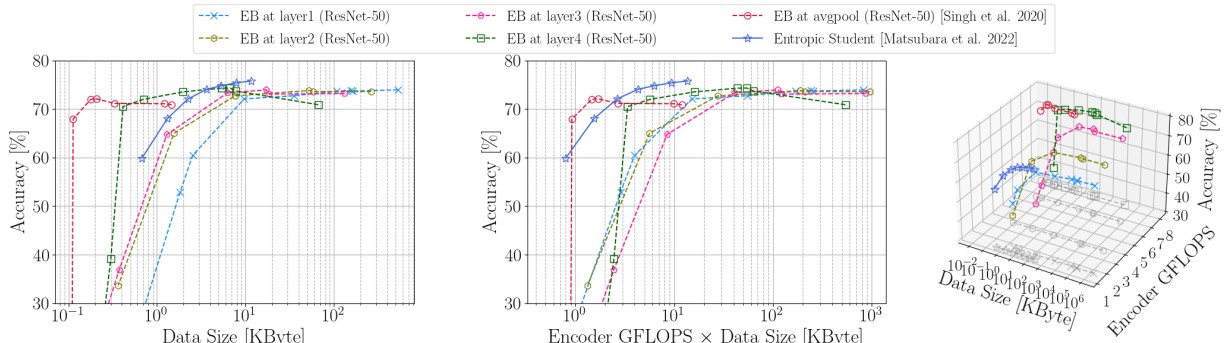

Figure 9: Entropic Student vs. ResNet-50 with entropy bottleneck (EB) introduced to different layers, using encoder FLOPs instead of encoder size (Eq. (1)) in Fig. 6 for our proposed ExR-D and three-way tradeoffs (**middle** and **right**). Note that the reported FLOPS do not cover all the operations in the encoders as PyTorch Profiler supports only specific operations such as matrix multiplication and 2D convolution at the time of writing.

we train the model for 60 epochs with batch size of 256.[9] Note that Singh et al. (2020) evaluate the accuracy of their models on a $299 \times 299$ center crop. Since the pretrained ResNet-50 expects the crop size of $224 \times 224$,[10] we use the crop size for all the considered classifiers to highlight the effectiveness of the training method.

$$\mathcal{L} = \underbrace{\mathcal{L}_{\text{cls}}(\hat{\mathbf{y}}, \mathbf{y})}_{\text{distortion}} - \beta \underbrace{\log p_\phi(f_\theta(\mathbf{x}) + \epsilon)}_{\text{rate}}, \quad \epsilon \sim \text{Unif}(-\tfrac{1}{2}, \tfrac{1}{2}) \tag{8}$$

### D.2.2 Object detection

Reusing the model trained on the ImageNet dataset with the end-to-end training method, we fine-tune Faster R-CNN (Ren et al., 2015). Since we empirically find that a simple transfer learning approach[11] to Faster R-CNN with the model trained by the baseline method did not converge, we use the 2nd stage of the fine-tuning method described above. The hyperparameters are the same as above, but the number of epochs for the 2nd stage training is 5.

### D.2.3 Semantic segmentation

We fine-tune DeepLabv3 (Chen et al., 2017b) with the same student model trained on the ImageNet dataset. Using the SGD optimizer with an initial learning rate of 0.0025, momentum of 0.9, and weight decay of $10^{-4}$, we minimize the cross entropy loss. The learning rate is adjusted by Eq. (7), and we train the model for 50 epochs with batch size of 8.

## E Encoder FLOPS

As mentioned in Section 3.1.3, we defined and introduced encoder size (Eq. (1)) as an additional metric for the ExR-D and three-way tradeoffs since FLOPS is not a static value, and FLOP/MAC is not well supported by existing PyTorch frameworks such as PyTorch Profiler[12], fvcore, and deepspeed. Figure 9 shows results of the bottleneck placement experiment (Fig. 6), using encoder FLOPS approximated by PyTorch Profiler instead of the encoder size. While its supported operations are limited (*e.g.*, matrix multiplication and 2D convolution at the time of writing), we confirmed similar trends in Fig. 9 to those in Fig. 6.

---

[9]For the ImageNet dataset, Singh et al. (2020) train their models for 300k steps with batch size of 256 for 1.28M training samples, which is equivalent to 60 epochs (= $\frac{300k \times 256}{1.28M}$).

[10]https://pytorch.org/vision/stable/models.html#classification

[11]https://github.com/pytorch/vision/tree/main/references/detection

[12]https://pytorch.org/docs/stable/profiler.html

