# OpenReview forum: "SC2 Benchmark: Supervised Compression for Split Computing"
_TMLR — Accepted by TMLR_

### Review · Reviewer_X4aF · 2023-03-11

**Summary Of Contributions:**

This paper discusses split computing for deep learning models between mobile devices and edge servers, which is a popular solution for implementing such models on mobile devices. The authors propose a new method called supervised compression for split computing (SC2) and new evaluation criteria to minimize computation on mobile devices, minimize transmitted data size, and maximize model accuracy. They conducted a comprehensive benchmark study and released a Python package to help researchers better understand the tradeoffs of supervised compression in split computing.

**Audience:**

Yes

**Claims And Evidence:**

Yes

**Requested Changes:**

Please refer to the weaknesses section for more details.

**Strengths And Weaknesses:**

---

**Strengths**:
* The paper is well-written and easy to understand. The authors provide sufficient background on split computing to make it accessible to readers outside the field.
* This paper addresses an important problem in the emerging field of split computing. By providing a standardized way to measure the performance of split computing systems, this benchmark enables researchers to compare different systems and evaluate their performance objectively. This helps promote the development of split computing and advance the field.

---

**Weaknesses**:
* The metrics used in this paper may be subject to some questions. Firstly, the encoder cost is measured by the model size, rather than the more commonly used metric of the number of Multiply-Accumulate operations (MACs). While the authors justify their choice, most papers in efficient deep learning use MACs as a proxy for computation cost. Secondly, the authors do not consider the cost of the decoder, even though its computation cost could affect the end-to-end latency. Although the decoder runs on a faster server, its cost should be taken into account. Thirdly, the proposed ExR metric seems arbitrary, and it is unclear why these two metrics should be multiplied. While it is important to use a proxy metric that is not hardware-specific, the authors should justify why their proposed proxy metric is indeed correlated with the end-to-end latency, which is the most important metric in this context. Ultimately, the end-to-end latency includes the inference time of the encoder, transmission time of the compressed data, and the inference time of the decoder, making it the most relevant metric.
* The benchmarks included in this paper are limited to image workloads. In order to make the benchmark more comprehensive, it is important to include other types of workloads such as speech recognition, machine translation, language modeling, and text summarization. By including a wider variety of workloads, the benchmark can provide a more complete picture of the performance of split computing systems across different tasks. This would be particularly beneficial for researchers and practitioners working in these other areas.

---

---

> ### Author Response · Authors · 2023-04-09
> **Response to Reviewer X4aF**
>
> > The paper is well-written and easy to understand. The authors provide sufficient background on split computing to make it accessible to readers outside the field.
>
> > This paper addresses an important problem in the emerging field of split computing. By providing a standardized way to measure the performance of split computing systems, this benchmark enables researchers to compare different systems and evaluate their performance objectively. This helps promote the development of split computing and advance the field.
>
> We thank the reviewer for their positive feedback and recognizing its impact on the community.
>
> ---
>
> > The metrics used in this paper may be subject to some questions. Firstly, the encoder cost is measured by the model size, rather than the more commonly used metric of the number of Multiply-Accumulate operations (MACs). While the authors justify their choice, most papers in efficient deep learning use MACs as a proxy for computation cost.
>
> In this work, we define and prefer encoder size as the total number of bits to represent the encoder for taking into account quantized models as well. MAC do not consider quantization, and tracing operations in various models (including quantized/binarized models) is an engineering challenge.
> **We also want to refer the reviewer to [our comment to all the reviewers](https://openreview.net/forum?id=p28wv4G65d&noteId=cEVbGyznbL) where we provide a simple, concrete example that even popular PyTorch frameworks cannot accurately estimate the MAC/FLOPS, for emphasizing why we intentionally did not use MAC/FLOPS. In the comment, we also kindly request the reviewer’s opinion.**
>
>
> ---
>
> > Secondly, the authors do not consider the cost of the decoder, even though its computation cost could affect the end-to-end latency. Although the decoder runs on a faster server, its cost should be taken into account.
>
> As empirically shown in existing studies (Matsubara et al., 2020, 2022c, Matsubara & Levorato 2021), the computing cost on the edge server is negligible for effective split computing approaches as the edge server in the split computing scenarios (see Fig. 1) has stronger computing resources than the mobile (local) device has.
>
>
>
> ---
>
> > Thirdly, the proposed ExR metric seems arbitrary, and it is unclear why these two metrics should be multiplied. While it is important to use a proxy metric that is not hardware-specific, the authors should justify why their proposed proxy metric is indeed correlated with the end-to-end latency, which is the most important metric in this context. Ultimately, the end-to-end latency includes the inference time of the encoder, transmission time of the compressed data, and the inference time of the decoder, making it the most relevant metric.
>
> As the reviewer correctly recognizes, the end-to-end latency evaluation results will depend on devices and network configurations, and we want to keep the metrics independent from hardware specifications.
> We stress that our main quantity of interest is the three-way tradeoff (3D plots) between rate, encoder model size, and prediction error. Since these three-way tradeoffs are difficult to display at times, we summarized encoder size and rate multiplicatively. We agree that this definition is a crude summarization, but we cannot additively summarize both quantities (which have different units and value ranges) without assuming an arbitrary weighting factor between the two.
>
> ---
>
> > The benchmarks included in this paper are limited to image workloads. In order to make the benchmark more comprehensive, it is important to include other types of workloads such as speech recognition, machine translation, language modeling, and text summarization. By including a wider variety of workloads, the benchmark can provide a more complete picture of the performance of split computing systems across different tasks. This would be particularly beneficial for researchers and practitioners working in these other areas.
>
>
> We emphasize that split computing is more beneficial for tasks that require relatively large input data at runtime, such as visual data. For applications involving text, the corresponding input data sizes are typically much smaller, making simply offloading the full computation to the edge server more efficient.
>
> For instance, NLP tasks use text inputs (e.g., string or sequence of token indices), which are already too small to be further compressed by dense representations. Thus transferring text data at runtime should not be a big problem.
>
> In our study, we use computer vision tasks since those are most popular in the recent split computing studies and use relatively larger data as input at runtime, which give us stronger motivations to discuss the SC2 benchmark.

---

> > ### Comment · Reviewer_X4aF · 2023-04-17
> > **Response**
> >
> > Thanks for the response! I understand that estimating the number of FLOPs accurately can be challenging due to the vague definition of what counts as a FLOP. However, the number of MACs is better defined as it only considers the multiply-accumulate operation. Therefore, the computation of z2 in the provided example should not be classified as a MAC. Although current tools may not provide accurate estimations, it is still a fair comparison as long as the tool uses a consistent definition, which is still a better indicator than the model size.

---

### Review · Reviewer_MSsY · 2023-04-01

**Summary Of Contributions:**

Split computing refers to a setup where a low-power edge device (e.g. mobile phone) does part of the computation for neural network inference, and a server does another part. For example, one may wish to classify an image captured on the device, but not run a large neural network there. If part of the computation is done on a server, some data has to be transmitted, and this (compressed) data can also be produced by a neural network (e.g. trained to preserve information useful for classification).

The present paper performs an extensive benchmark study to evaluate various design choices for split computing and their effect on supervised performance, compression rate, and low computational cost. Specifically, it looks at different (classical and learned) compression methods for inputs or features, supervised compression models, different tasks, architectures, and training schemes. The best performing model was found to be a neural image compression model trained by knowledge distillation.

**Audience:**

Yes

**Broader Impact Concerns:**

Split computing seems useful in warfare (e.g. autonomous drones) and mass surveillance, so an impact statement might be appropriate. This being a benchmark study, it is not clear that by itself it will make a difference though.

**Claims And Evidence:**

Yes

**Requested Changes:**

It would be nice to discuss the relation to the well-known information bottleneck, where a supervised loss is combined with a hidden-layer compression loss. The IB literature is a bit more theoretical, but it would be good to make the connection.

Typos:
page 4: "growing interest from the research."

**Strengths And Weaknesses:**

+ The paper is well written
+ Extensive and systematic experiments evalating various relevant design choices
+ Re-useable benchmark for future work

I did not find significant issues

---

> ### Author Response · Authors · 2023-04-09
> **Response to Reviewer MSsY**
>
> > The paper is well written
> > - Extensive and systematic experiments evalating various relevant design choices
> > - Re-useable benchmark for future work
> > - I did not find significant issues
>
> We thank the reviewer for their positive comments on our benchmark framework and study.
>
> ---
>
> > It would be nice to discuss the relation to the well-known information bottleneck, where a supervised loss is combined with a hidden-layer compression loss. The IB literature is a bit more theoretical, but it would be good to make the connection.
>
> Appendix D provides connections. In the revision, we also explicitly mention the connection in Section 2.2 (Supervised Compression) and explain Singh et al., 2021 as the first work that proposes an instantiation of information bottleneck for supervised compression.
>
> ---
>
> > Typos: page 4: "growing interest from the research."
>
> We thank the reviewer for carefully reading our paper. We fixed the typo in the revision.

---

### Review · Reviewer_Ley7 · 2023-04-06

**Summary Of Contributions:**

This study provides a comprehensive benchmark for supervised compression for split computing (SC2). Instead of only focusing on the traditional R-D tradeoff, this paper states that the size of encoder should also be considered, and hence proposes another tradeoff that takes encoder size and data size multiplicatively into account. Based on the proposed metric, the authors conduct a comprehensive benchmark study involving 10 different state of the art methods and more than 180 trained models for three computer vision tasks, such as image classification, object detection, and segmentation.

**Audience:**

Yes

**Claims And Evidence:**

Yes

**Requested Changes:**

Overall, I believe this paper provides valuable benefits to community and can be accepted if the above weaknesses can be addresses.

**Strengths And Weaknesses:**

Strengthens:
(1)	The paper is well-motivated. The rate-distortion evaluation metric is not sufficient to evaluate the performance-efficiency trade-off of SC2, given the prevailing of large-scale encoders used in SC2. It is timely to take the size of encoder into account.
(2)	The reason for the urgent requirement of a comprehensive benchmark listed in 2.3.3 is convincing and reasonable.
(3)	Figure 2 well explains the difference between SC and IC.
(4)	The empirical results look solid and interesting. Evaluation 10 baselines in one paper requires massive effort and time, which I highly appreciate it.

Weaknesses:
(1)	While I fully understand evaluating 10 baselines requires massive effort, it is important to report the details of hyperparameters and configurations used to reproduce the results of each baseline to ensure the fairness.
(2)	I have a big concern about the usage of parameter count as a proxy for the encoder size. While it is important to measure the parameter count, parameter count itself can not faithfully reflect the real speed of the model/algorithm on hardware. I encourage the authors also report the FLOPs of all the baselines. As long as we guarantee all the numbers are measured using the same hardware, it is a solid contribution and will provide a full picture to the readers.
(3)	What is the difference between EB at layer1 and Entropic Student in Figure 6? What causes the latter performs much better than the former?

---

> ### Author Response · Authors · 2023-04-09
> **Response to Reviewer Ley7**
>
> > Strengthens: (1) The paper is well-motivated. The rate-distortion evaluation metric is not sufficient to evaluate the performance-efficiency trade-off of SC2, given the prevailing of large-scale encoders used in SC2. It is timely to take the size of encoder into account. (2) The reason for the urgent requirement of a comprehensive benchmark listed in 2.3.3 is convincing and reasonable. (3) Figure 2 well explains the difference between SC and IC. (4) The empirical results look solid and interesting. Evaluation 10 baselines in one paper requires massive effort and time, which I highly appreciate it.
>
> We thank the reviewer for the very positive feedback and understanding key motivations and findings of our work.
>
> ---
>
> >  (1) While I fully understand evaluating 10 baselines requires massive effort, it is important to report the details of hyperparameters and configurations used to reproduce the results of each baseline to ensure the fairness
>
> We agree that it is important to provide hyperparameters and configurations.
> In fact,
> 1. Appendices A - D describe the complete hyperparameters and configurations, and
> 2. our supplementary material contains the configurations used to obtain the results in this work, which will be published as a code repository upon acceptance.
>
> ---
>
> >  (2) I have a big concern about the usage of parameter count as a proxy for the encoder size. While it is important to measure the parameter count, parameter count itself can not faithfully reflect the real speed of the model/algorithm on hardware. I encourage the authors also report the FLOPs of all the baselines. As long as we guarantee all the numbers are measured using the same hardware, it is a solid contribution and will provide a full picture to the readers.
>
> We defined encoder size as the total number of bits needed to represent the parameters, which is not just parameter count, but considers types of quantized data e.g., int8-quantized encoder is better than float32 encoder if both the encoders have the same network architecture.
> As described in Section 3.1.3, we intentionally did not use FLOPS since it is not well-defined and is difficult to accurately compute the metric for arbitrary modules/layers/models implemented with deep learning frameworks (PyTorch in this study) e.g., detectron2 (fvcore) leaves [a statement “Flop is not a well-defined concept.”](https://detectron2.readthedocs.io/en/latest/modules/fvcore.html#fvcore.nn.FlopCountAnalysis)
>
>
> **We also want to refer the reviewer to [our comment to all the reviewers](https://openreview.net/forum?id=p28wv4G65d&noteId=cEVbGyznbL) where we provide a simple, concrete example that even popular PyTorch frameworks cannot accurately estimate the MAC/FLOPS, for emphasizing why we intentionally did not use MAC/FLOPS. In the comment, we also kindly request the reviewer’s opinion.**
>
> ---
>
> > (3) What is the difference between EB at layer1 and Entropic Student in Figure 6? What causes the latter performs much better than the former?
>
> There are two main differences between a model with EB introduced at layer1 and the Entropic Student.
> 1. The former approach introduces EB to a reference model (ResNet-50 in Figure 6) and then trains the extended reference model, whose architecture is not necessarily designed to perform well in split computing scenarios. The latter approach replaces the first layers in the same reference model with a neural compression encoder-decoder designed for split computing.
> 2. Entropic Student also leverages a pretrained reference model as a teacher model for the two-stage training. As shown in both Figure 3 and (Matsubara et al. 2022), the training method improved the performance compared to training exactly the same model architecture without knowledge distillation.
>
> ---
>
> > Overall, I believe this paper provides valuable benefits to community and can be accepted if the above weaknesses can be addresses.
>
> We appreciate the reviewer’s positive comments on contribution to the community. We believe that our responses address each of the weaknesses above.

---

### Review · Reviewer_Y1Q3 · 2023-04-08

**Summary Of Contributions:**

In this paper, the authors explore the concept of supervised compression for split computing, a valuable approach when an edge device and an edge server collaborate to execute tasks such as image classification and object detection. They thoroughly investigate various techniques for implementing split computing, taking into account three key metrics: encoder size, data size, and supervised distortion.

To further support research in this area, the authors introduce a novel benchmark known as the SC2 Benchmark, specifically designed to evaluate the effectiveness of different split computing methodologies. Through their comprehensive analysis, they demonstrate that the neural image compression model outperforms other approaches in terms of supervised rate-distortion performance.

**Audience:**

Yes

**Claims And Evidence:**

Yes

**Requested Changes:**

Please refer to the weakness part mentioned above.

**Strengths And Weaknesses:**

[Strength]
- This paper is not only well-written but also easy to follow, providing a clear and engaging discussion of the topic.
- It offers a thorough description of various previous techniques related to split computing, which adds depth to the analysis.
- The introduction of the benchmark will prove invaluable for researchers seeking to conduct fair and accurate comparisons in the field of split computing.

[Weakness]
- The paper could benefit from a more concise presentation of previous methods, as some explanations appear to be repetitive across sections.
- The models and tasks discussed may seem outdated, and incorporating more recent advancements would strengthen the paper's relevance.
- Given the varying characteristics of split computing, depending on scenarios and advancements in edge devices or servers, it would be helpful if the authors could discuss whether Equation (1) is the most appropriate hardware-agnostic metric. A metric that takes power or hardware cost of edge devices into account might be more challenging to define, but it could be essential for evaluating split computing methodologies.
- It would be valuable to explore whether the neural image compression model remains the best approach for other tasks, or if the conclusions drawn are specific to certain tasks such as image classification or object detection.
- The overall contributions of this paper appear to be incremental, and the work presented resembles a technical report rather than a novel research paper. To enhance its impact, the authors could consider incorporating more innovative ideas or findings.

---

> ### Author Response · Authors · 2023-04-09
> **Response to Reviewer Y1Q3**
>
> > [Strength]
> > - This paper is not only well-written but also easy to follow, providing a clear and engaging discussion of the topic.
> > - It offers a thorough description of various previous techniques related to split computing, which adds depth to the analysis.
> > - The introduction of the benchmark will prove invaluable for researchers seeking to conduct fair and accurate comparisons in the field of split computing.
>
> We thank the reviewer for the positive feedback, especially on how valuable this benchmark will be for the research community.
>
> ---
>
>
> > [Weakness]
>
> > The paper could benefit from a more concise presentation of previous methods, as some explanations appear to be repetitive across sections.
>
> It would be appreciated if the reviewer specifies which sections/paragraphs they found repetitive. We provide the details of previous methods (models) in Appendices A - D to concisely present the previous methods in the main content.
>
> ---
>
> > The models and tasks discussed may seem outdated, and incorporating more recent advancements would strengthen the paper's relevance.
>
> We respectfully disagree with the subjective assessment.
> Supervised compression is a relatively new problem (defined in 2022), and our study discusses recent models and methods such as RegNet (Radosavovic et al., 2020), Hybrid ViT (Steiner et al., 2021), CR+BQ (Matsubara & Levorato, 2021), Codec-based feature compression (Alvar & Bajic, 2021), Compressive Feature (Singh et al., 2020; Yuan et al., 2022), and Entropic Student (Matsubara et al., 2022c).
>
> ---
>
> > Given the varying characteristics of split computing, depending on scenarios and advancements in edge devices or servers, it would be helpful if the authors could discuss whether Equation (1) is the most appropriate hardware-agnostic metric. A metric that takes power or hardware cost of edge devices into account might be more challenging to define, but it could be essential for evaluating split computing methodologies.
>
> As mentioned in Section 3.1.3, we also recognize MAC/FLOPS as alternative metrics, and MAC is a hardware-agnostic metric. However, it is extremely challenging to accurately measure MAC/FLOPS of arbitrary PyTorch modules in a systematic way, and this is an engineering problem that the community has been struggling with for years. For these reasons, we prefer the encoder size (Eq. 1), which is a static value and can be accurately estimated independent from hardware devices and considers any learnable modules/layers.
>
> **We also want to refer the reviewer to [our comment to all the reviewers](https://openreview.net/forum?id=p28wv4G65d&noteId=cEVbGyznbL) where we provide a simple, concrete example that even popular PyTorch frameworks cannot accurately estimate the MAC/FLOPS, for emphasizing why we intentionally did not use MAC/FLOPS. In the comment, we also kindly request the reviewer’s opinion.**
>
> ---
>
> > It would be valuable to explore whether the neural image compression model remains the best approach for other tasks, or if the conclusions drawn are specific to certain tasks such as image classification or object detection.
>
> Our claims in this study are based on the results for the three supervised tasks: image classification, object detection, and semantic segmentation. In the revision, we clarified this point in Section 5 (Conclusion).
>
> ---
>
> > The overall contributions of this paper appear to be incremental, and the work presented resembles a technical report rather than a novel research paper. To enhance its impact, the authors could consider incorporating more innovative ideas or findings.
>
> While we respectfully disagree with the subjective assessment, **we want to refer the reviewer to [TMLR’s acceptance criteria](https://www.jmlr.org/tmlr/acceptance-criteria.html) as this type of the assessment is not aligned with TMLR’s criteria**.
>
> ```
> ...
> Crucially, it should not be used as a reason to reject work that isn't considered “significant” or “impactful” because it isn't achieving a new state-of-the-art on some benchmark. Nor should it form the basis for rejecting work on a method considered not “novel enough”, as novelty of the studied method is not a necessary criteria for acceptance. We explicitly avoid these terms (“significant”, “impactful”, “novel”), and focus instead on the notion of “interest”. If the authors make it clear that there is something to be learned by some researchers in their area from their work, then the criteria of interest is considered satisfied. TMLR instead relies on certifications (such as “Featured” and “Outstanding”) to provide annotations on submissions that pertain to (more speculative) assertions on significance or potential for impact.
> ...
> ```

---

### Author Response · Authors · 2023-04-09
**Common response to all the reviewers; why we prefer encoder size to MAC/FLOPS of encoder**

Dear all the reviewers,

We thank the reviewers for reviewing our paper.
We are leaving this comment as we confirmed that some of the reviewers have similar questions, why we used the encoder size defined in Eq. (1)  instead of MAC/FLOPS of encoder.


First of all, **we are open to replacing the encoder size in Eq. (1) with MAC or FLOPS if the reviewers believe that MAC or FLOPS should outweigh the advantages of the encoder size, after considering the issues of the metrics provided below**.

In short, it is very difficult to accurately estimate MAC/FLOPS of arbitrary models implemented with deep learning frameworks such as PyTorch. **The encoder size we defined in Eq. (1) is a hardware-agnostic metric that can be accurately estimated in a systematic way and considers 1) any learnable modules/layers and 2) types of quantized data.**

To highlight this issue of MAC/FLOPS, we provide a simple example.

```python
from fvcore.nn import FlopCountAnalysis
from torch import nn
from deepspeed.profiling.flops_profiler.profiler import FlopsProfiler
import torch

class DummyEncoder(nn.Module):
    def __init__(self):
        super().__init__()
        self.conv1 = nn.Conv2d(in_channels=3, out_channels=1, kernel_size=1)
        self.bias = nn.Parameter(torch.rand((3, 20, 20)))
        self.conv2 = nn.Conv2d(in_channels=3, out_channels=1, kernel_size=1)

    def forward(self, x):
            z1 = self.conv1(x)
            z2 = x + self.bias
            z3 = self.conv2(z2)
            z4 = z1 * z3
            return z4

encoder = DummyEncoder()
x = torch.rand(1, 3, 20, 20)

# fvcore
print('[ fvcore ]')
macs = FlopCountAnalysis(encoder, x)
# Note: fvcore.nn.FlopCountAnalysis counts one fused multiply-add as one flop
print(macs.total()) # shows 2400
print(macs.total('conv1')) # shows 1200
print(macs.total('conv2')) # shows 1200
# macs.total() == macs.total('conv1') + macs.total('conv2')
# which means that it misses MACs to compute z2 and z4


def get_macs_by_deepspeed(module, name):
    prof = FlopsProfiler(module)
    prof.start_profile()
    z = module(x)
    print(name, ':', prof.get_total_macs())
    prof.end_profile()

# deepspeed (might need a GPU and NVIDIA driver to run FlopsProfiler)
print('[ deepspeed ]')
get_macs_by_deepspeed(encoder, 'encoder') # shows 2400
get_macs_by_deepspeed(encoder.conv1, 'encoder.conv1') # shows 1200
get_macs_by_deepspeed(encoder.conv2, 'encoder.conv2') # shows 1200
# get_macs_by_deepspeed(encoder, 'encoder') == get_macs_by_deepspeed(encoder.conv1, 'encoder.conv1') + get_macs_by_deepspeed(encoder.conv2, 'encoder.conv2')
# which means that it misses MACs to compute z2 and z4

```

This sample encoder takes x as input, say x is a tensor in shape of (1, 3, 20, 20) and uses 2 convolution layers and 1 bias term in shape of (3, 20, 20).
Both the PyTorch tools
- return 2400 as the total MAC of `encoder` (sum of two MACs to compute z1 and z3) and
- miss MACs to compute z2 and z4.


As of today, popular PyTorch tools such as fvcore from Meta (mentioned in the manuscript) and deepspeed from Microsoft cannot trace the operations to compute either z2 or z4 to measure MAC/FLOPS. Both of these frameworks estimate MAC/FLOPS of `DummyEncoder` as the sum of those for z1 and z3 only, because z2 and z4 are computed by low-level operations.
Those low-level operations are frequently used in deep learning model implementations (e.g., “shortcut” in ResNet is a well-known example) including those used in neural compression packages.

Furthermore, the measured numbers in the above example do not take into account the data types (e.g., int8, float16, float32) while our encoder size metric does.

In summary, we acknowledge that many of the efficient deep learning papers use MAC.
However, there remains an engineering issue in the community that it is extremely challenging to accurately measure MAC/FLOPS of arbitrary PyTorch modules in a systematic way. As shown in the above example, the existing frameworks cannot systematically trace all the operations even for such a simple model. For these reasons, we defined encoder size in Eq. (1), which is a static value that can be accurately estimated independent from hardware devices and considers any learnable modules/layers. For lowering the barrier to SC2 studies, it should also be important that the benchmark uses metrics that the community can compute easily and accurately.

**We kindly request opinions from the reviewers during the period of interaction and discussion, whether or not MAC or FLOPS should outweigh the advantages of the encoder size (hardware-agnostic, considering all learnable modules/layers and types of quantized data), so that we can reflect the opinions for the revision.**

We are looking forward to discussing it with the reviewers.
Sincerely,
Authors

---

### Comment · Action_Editors · 2023-04-17
**Hi, all**

Dear Reviewers,

Thanks for the great efforts in reviewing the paper. Since authours have provided some explanations in the rebuttal, it would be much appreciated if you can help  check the rebuttal of authours. Thanks!

best,

AE

---

### Decision · Action_Editors · 2023-05-25

**Recommendation:** Accept as is

**Comment:**

Please refer to my comments above.

**Audience:**

This paper discussed an interesting topic -- supervised compression for split computing (SC2). So the researchers who are working on machine learning, computing, and even hard-ware related topics will be interested in this paper.

**Claims And Evidence:**

This paper is reviewed by four experts. Three of them are positive, while the other reviewer concerns about the readership of this paper.

In general, all the reviewers agree with the merits of this paper which gives a comprehensive benchmark for supervised compression for split computing (SC2).  It is a valuable approach when an edge device and an edge server collaborate to execute tasks such as image classification and object detection. The authours made extensive efforts on the evaluation.The claims of this paper are well supported by the experiments. The initial comments asked for some experimental  details such as configurations, hyper-parameters, and metrics.  The authours explicitly explain these details.

Given these points above, the AE suggests the acceptance given the acceptance of this paper.

---

> ### Author Response · Authors · 2023-06-14
> **Thank you for reviewing our paper**
>
> We would like to thank the action editor and the anonymous reviewers for their time and feedback.
>
> We just uploaded the recorded presentation and camera-ready, which includes a new appendix to present additional evaluation results with encoder FLOPS instead of encoder size as we promised.